# From insufficient rainfall to livelihoods: understanding the cascade of drought impacts and policy implications

Louise Cavalcante[1], David W. Walker[2], Sarra Kchouk[2], Germano Ribeiro Neto[3], Taís Maria Nunes Carvalho[4], Mariana Madruga de Brito[4], Wieke Pot[1], Art Dewulf[1], Pieter R. van Oel[2]

[1] Public Administration and Policy Group, Wageningen University, Wageningen, The Netherlands.

[2] Water Resources Management Group, Wageningen University, Wageningen, The Netherlands.

[3]Hydrology and Quantitative Water Management Group, Wageningen University, Wageningen, The Netherlands.

[4]Department of Urban and Environmental Sociology, UFZ-Helmholtz Centre for Environmental Research, Leipzig, Germany

*Correspondence to*: louise.cavalcantedesouzacabral@wur.nl

## Abstract

A cascade of drought impacts refers to a series of interconnected events that trigger a chain reaction of impacts, extending beyond water scarcity, to affect agricultural production, socio-economic factors, and the environment. This paper aims to understand the role of society in mitigating drought impacts, particularly through policy responses. Conducting a case study in Ceará state, northeast Brazil, we used a global rare dataset of continuously drought monitoring, complemented by interviews with smallholder farmers and agricultural extension technicians. Additionally, we analyzed policy documents related to public policies implemented at the local level. Employing a classification of drought impacts as our analytical framework, our findings indicate that socio-environmental-economic impacts of drought are less frequently reported, suggesting that development policies are mitigating cascading effects on livelihoods. Most impacts are associated with hydrological impacts of drought, suggesting unintended consequences of investments in increasing water supply. We emphasize the significant contribution of public policies to mitigating the cascading effects of drought, which do not necessarily involve increasing water availability, but strengthen the local economy.

# 1. Introduction

The Integrated Drought Management Programme (IDMP)'s guidelines for developing national drought policy begin with a preface by Michel Jarraud, the then secretary-general of the World Meteorological Organization, who stated in 2013: "Both at the national and regional scale, responses [to drought] are known to be often untimely, poorly coordinated and lacking the necessary integration. As a result, the economic, social and environmental impacts of droughts have increased significantly in many regions of the world. We simply cannot afford to continue in a piecemeal mode, driven by crisis rather than prevention. We have the knowledge, we have the experience, and we can reduce the impacts of droughts. What we need now is a policy framework and action on the ground for all countries that suffer from droughts. Without coordinated national drought policies, nations will continue to respond to drought in a reactive way" (WMO & GWP, 2014). Many countries that regularly experience droughts now have both dedicated drought policies and other policies designed to increase resilience and reduce drought impacts. But how do we know if these policies are working? This study analyzed a continuously monitored drought impacts dataset, stakeholder interviews, and policy documents to assess if we have progressed since Michel Jarraud's statement, and drought policies are now coordinated, integrated and focused on prevention rather than reaction.

Due to the complexity of drought, scholars are continuously engaging with and stay informed about the latest discussions and advancements in the subject because there is no universal definition of drought. This ongoing engagement highlights the multidisciplinary interest in the subject (Mishra & Singh, 2010; Lloyd-Hughes, 2014; Wilhite & Glantz, 1985). In the context of climate change, defining drought becomes even more challenging, as it is difficult to establish climatological norms for the various components of the local water balance. As human activities increasingly impact the environment, there is a growing need for an integrated approach that considers both natural and human factors. Recent research suggests that drought should be viewed and understood as a process, not merely a product. It involves complex interactions between natural and human-induced changes, such as climate change, land and water management, and human decision-making (AghaKouchak et al., 2021).

Different categories of drought are understood based on the specific context and disciplinary perspectives through which they are examined. For example, meteorologists might define drought in terms of precipitation deficits, focusing on meteorological drought—characterized by prolonged periods of insufficient precipitation, often coupled with increased evapotranspiration, affecting large geographic areas (Wilhite et al., 1985). Agricultural scientists, on the other hand, might emphasize soil moisture levels and the impact on crops, leading to a focus on agricultural drought, which occurs when a lack of soil moisture prevents plants from growing, often due to precipitation shortages and/or high evapotranspiration rates (Wilhite et al., 1985). Hydrologists typically concentrate on the availability of surface and groundwater resources, categorizing drought from a hydrological perspective, which includes negative anomalies in surface and groundwater, such as below-normal groundwater levels, reduced water levels in lakes, shrinking wetlands, and

diminished river discharge (Van Loon, 2015). Another category, often considered by ecologists, is
environmental or ecosystem drought, which refers to a temporary shortfall in water availability that
pushes ecosystems beyond their vulnerability limits, disrupts ecosystem services, and triggers
feedback loops within both natural and human systems (Crausbay et al., 2017).
When attempting to describe the social components intertwined with complex interactions, such as
those found in socioeconomic drought, important questions arise about where the physical aspects
of drought end and the human impacts begin. Socioeconomic drought has traditionally been linked
to the imbalance between water supply and societal water demands (Wilhite & Glantz, 1985).
However, this type of drought is not merely about the physical scarcity of water but rather the
broader societal and economic consequences that arise from it. While recent reflections have
expanded the concept of socioeconomic drought to include indirect impacts beyond just the lack
of water (Kchouk et al., 2023), many still rely on indices based on physical data to assess these
droughts. For instance, indices like the Water Resources System Resilience Index (WRSRI) are
used to more accurately identify the onset and duration of socioeconomic drought events (SEDEs)
(Wang et al., 2023b). The transition from meteorological and hydrological drought to
socioeconomic drought has been analyzed using linear methods (Wang et al., 2023a). However, a
significant limitation of these approaches is the absence of direct social data, such as the impacts
on populations, economic activities, social vulnerability, or public response to drought conditions.
Each of these drought types is closely intertwined with different societal impacts. For instance,
hydrological drought may lead to diminished water availability for human and animal
consumption, irrigation and industrial purposes. Agricultural drought is distinctly associated with
crop development impacts. Socioeconomic droughts impact people's lives, ecosystems, and
economic activities. Meteorological drought is a key driver for all other drought impacts (Van Loon
et al., 2016; Mishra and Singh, 2010). In this paper, we use these different types of drought impacts
as an analytical framework by categorizing and evaluating the diverse impacts associated with each
type of drought.
Although this classification is useful for presenting the results, we are in line with recent arguments
that drought should not be perceived as an isolated event, but as a continuous and interconnected
phenomenon that evolves over time. Moreover, drought impacts cascade through society and
economy at different speeds, affecting various groups and regions with varying intensities and
timings, potentially far from where the drought originated (Van Loon et al., n.d.). Therefore, there
is a need for a comprehensive understanding of the compound and cascading impacts of droughts
by considering interconnected natural and social systems, and the complex interactions between
different sectors affected by the impacts (de Brito et al., 2024).
We take the approach in which physical and social impacts are closely interconnected, and drought
impacts can cascade, in which one impact is connected to another, forming a chain reaction of
impacts (de Brito, 2021). For example, insufficient rainfall results in low soil moisture, leading to
reduced crop development, which in turn yields reduced harvests. This translates to diminished
earnings for the farmer, which contributes to higher food prices due to shortages, ultimately
culminating in heightened food insecurity. Despite the consequences of these cascading impacts,
we still have limited understanding of the relationships between them. Furthermore, research on
the effects of response measures on the attenuation or exacerbation of cascading impacts is scarce
(de Brito et al., 2024). To address these gaps here we focus on the societal aspects of drought
impacts, a significant dimension often overlooked in drought monitoring, which traditionally
concentrates more on the hydrometeorological drivers of these impacts (Kchouk et al., 2022).
Progress has been made in understanding the human impact on drought aggravation, such as the
influence of reservoirs on hydrological processes (Ribeiro Neto et al., 2022; Ribeiro Neto et al.,
2024) and groundwater depletion due to abstraction (Apurv et al., 2017). Yet there remains a
notable gap in understanding the societal role in mitigating drought, which could be tackled by
including social sciences to capture the complexity of relationships of society and the environment
in drought research (Kchouk et al., 2022; Savelli et al., 2022; Walker et al., 2022; Ribeiro Neto et
al., 2023). Here, we expand on our previous study, which highlighted the importance of monitoring
drought impacts in assessing drought, and advocates for ongoing monitoring of impacts (Walker
et al., 2024). We take a step further in this by approximating policy sciences to drought
management by generating knowledge from the following research question: *How do drought*
*impacts cascade and how do policy responses evolve to alleviate the impacts*? This paper aims to
understand the role of society in mitigating drought impacts, particularly through policy responses.
Through this research, we will explore the intersection of drought management and policy sciences
by generating insights into the role of public policies in alleviating the impacts of drought.
To explore this question, we consider Ceará, northeast Brazil, because it is the most advanced state
in Brazil with the implementation of the Drought Monitor, with the final step of incorporation of
local on-the-ground impacts data from agricultural extension technicians hereinafter referred to as
*observers*. Our analysis integrates three distinct qualitative datasets. The first dataset is a globally
rare example of spatially distributed, continual impacts monitoring conducted by observers, who
provide agricultural assistance to farmers. The second comprises information gathered through
interviews of smallholder farmers and observers during fieldwork. The third consists of policy
documents related to public policies implemented in the region.
In this study, we leverage data from traditionally low-data environments, recognizing the
significance of these often-overlooked sources as a valuable epistemic contribution to the study of
droughts. We integrate and validate these datasets to demonstrate their critical role in enhancing
our understanding of drought dynamics, particularly in regions that are among the most vulnerable
to drought, yet lack robust on the ground information. Our focus is specifically on the impacts on
smallholders, commonly referred to as family agriculture in Brazil, as they represent one of the
most vulnerable groups to the effects of climate extremes.

## 2. Methodology

This research constitutes a case study investigating the role of policy responses in alleviating the
impacts of drought. Specifically, we delve into the context of Ceará state to obtain insights by
examining the effects of particular policies implemented in the region on mitigating drought
impacts.

## 2.1 Study Area

The map illustrates the geographic layout of Ceará State, located in Northeast Brazil, which covers
a total area of approximately 148,920 square kilometers. The state is divided into 184
municipalities, as outlined by the purple boundaries on the map. These municipalities are home to
around 9 million people. In addition to the municipal boundaries, the map highlights the semiarid
region of Ceará with a light orange shading. This semiarid delimitation is significant as it
encompasses areas that are particularly susceptible to droughts and related environmental
challenges. The semiarid region covers a substantial portion of the state, influencing both the
climate and the socio-economic conditions of the municipalities within this area Figure 1.
The state has various economic activities, with key sectors including industry, particularly textiles
and automotive manufacturing, as well as tourism driven by its tropical beaches and wind sports.
Additionally, agriculture plays a crucial role, featuring crops like sugarcane, corn, beans, and fruits.
This agricultural sector, particularly family farming, that is most impacted by drought, affecting
both medium to large farmers and smallholder farmers (Pereira and Cuellar, 2015).

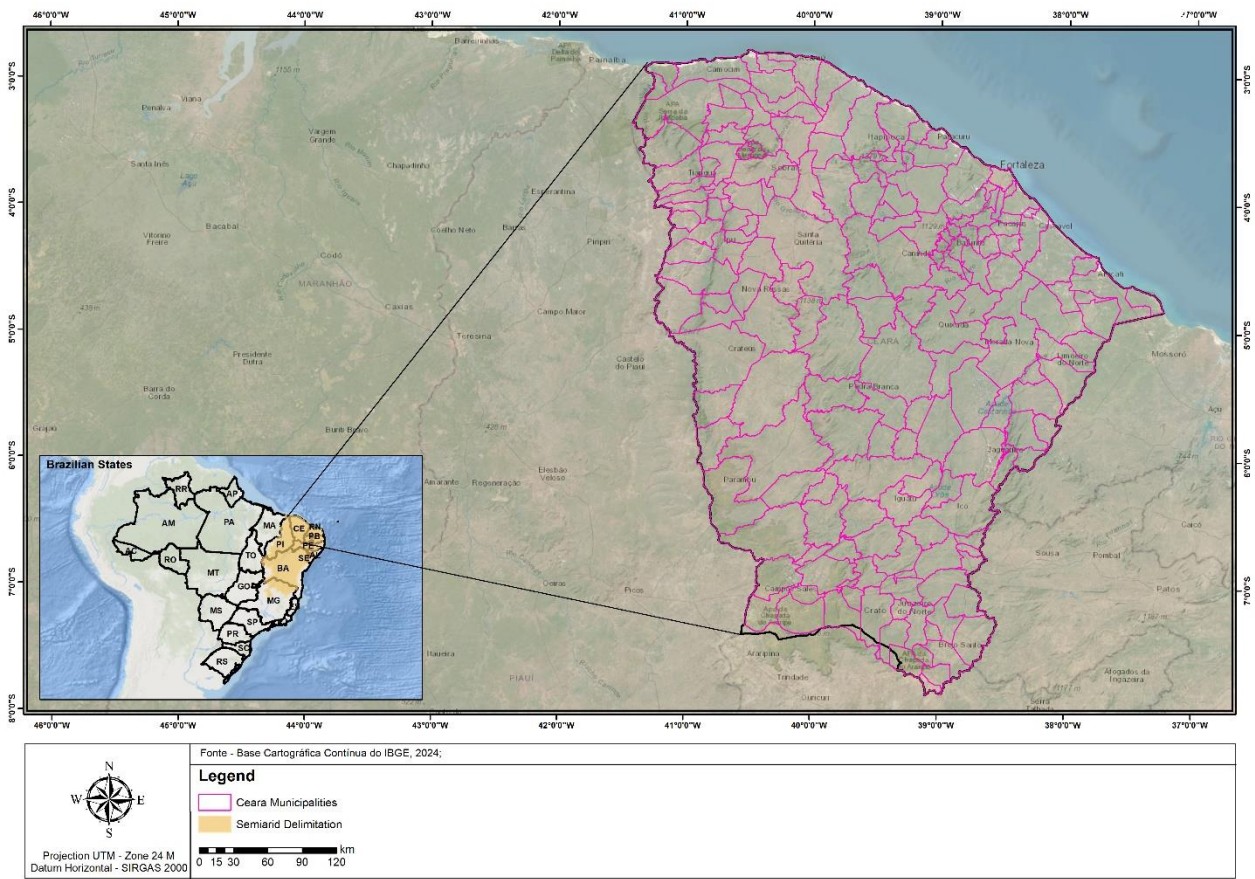

Figure 1: Geographic overview of Ceará State, Brazil.
The region is semiarid with a history of drought events (for more details on drought in the region,
see: (Marengo et al., 2017)). The most recent multi-year drought event (2012-2018) affecting the
region led to a state of emergency and impacted the economic growth of municipalities (De
Oliveira, 2019). During this drought, the Drought Monitor was implemented after a lengthy period
of political and technical negotiations (for more information, see (Cavalcante et al., 2023; Gutiérrez
et al., 2014)), starting in 2014 with the northeast region, and has since incrementally expanded to
cover all Brazilian states. Within the context of this tool, a map of drought severity is elaborated
and published online every month at https://monitordesecas.ana.gov.br/. Overall, it is a tool to
support dialogue between states and the federal government in addressing drought risks and
conditions and drought preparedness planning.
The mapping process for drought involves integrating relevant regional meteorological databases
and remote sensing analyses to compute indices. Validation follows ground observations of
drought impacts from networks of observers. This collaborative effort aims to ground truth in the
Drought Monitor using monthly questionnaires completed per municipality. It was initiated in
Ceará in 2019 by the state government rural/agricultural extension service (in Portuguese, *Empresa*
*de Assistência Técnica e Extensão Rural do Ceará)* for now on referred to as Ematerce. The data
collected validates the mapping process and contributes to refining drought monitoring systems.
Although the research had a weakness in starting monitoring during a non-drought period, it still
provided insightful findings on the effects of drought in the area, even in years that were not
considered statistically dry.
**2.2 Study data**
This research uses a range of qualitative data asynchronously gathered in a multi-step approach.
We used three qualitative datasets (1) obtained from an innovative drought monitoring instrument
in Brazil; (2) interviews with smallholder farmers and observers during fieldwork in Ceará state;
and (3) policy documents on public policies implemented in the region (Table 1).
Table 1: Summary of datasets used in this research

| Dataset | Period of data collection | Scale |
|---|---|---|
| Drought impacts monitoring data | February 2019 to October 2022 | Municipality level |
| Field work data | July 2019, November and December 2021 April 2022 | Household level |
| Policy documents data | Not applicable | National level and State level |


The data collection process for this study was designed to capture the multifaceted impacts of
drought in the region. We employed a comprehensive approach by triangulating multiple data
sources and methodologies to capture a holistic understanding of the phenomenon. Figure 2

presents a workflow outlining the sequential steps involved, each corresponding to a specific dataset. For a more comprehensive overview with steps and codes, we refer readers to the supplement materials section S.1., and to another study using the drought monitoring data set (Walker et al., 2024).

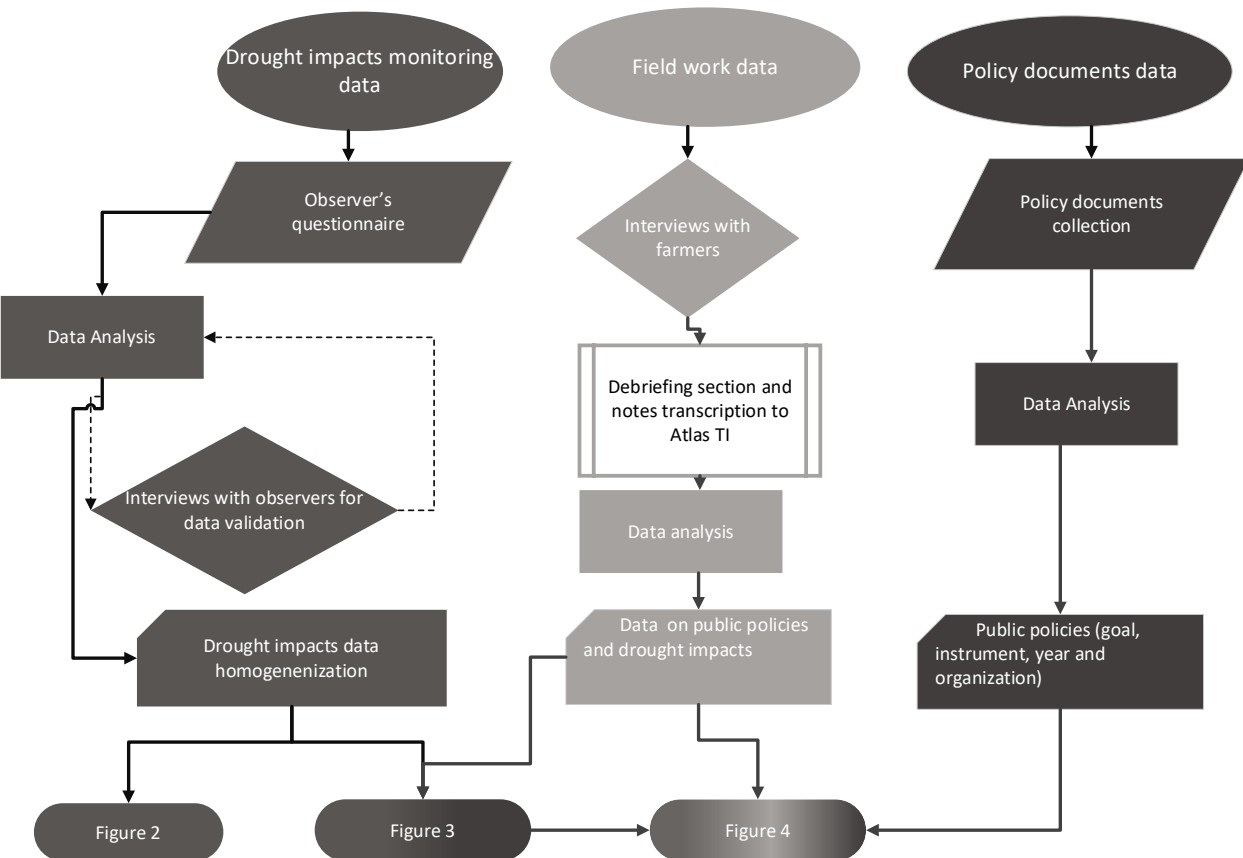

Figure 2: Summary of data collection and analysis methodology

## 2.2.1. Drought impacts monitoring data

Observers collected the first dataset as part of their job routine. In addition to their various tasks, they regularly complete monthly questionnaires for each municipality, providing information on drought impacts and other relevant information. Employed by Ematerce, these observers are based across the state, with most offices overseeing two or three municipalities, covering 184 municipalities. Thematic analysis was conducted on the responses to identify recurring patterns and themes. This type of analysis is particularly suitable for areas lacking empirical research and provides a rich description of predominant themes across the dataset (Braun & Clarke, 2006). Data validation interviews were conducted with observers to ensure the accuracy and comprehensiveness of the reported information.

In the homogenization step, we were interested in finding a common terminology to unify the understanding of local impacts observations, to achieve simplicity and manageability of data, to add clarity and focus on the most common patterns, and to increase readability. The outcome of

this effort was the identification of 14 distinct impact types (Table 3 in the Supplementary material), which were then classified into impacts due to drought impacts classification, i.e. hydrological, agricultural, and socio-environmental-economic impacts of drought.

**2.2.2. Fieldwork data**

Interviews and/or casual conversations were conducted with 60 smallholder farmers across multiple visits to the study area. Questions were formulated to encourage participants to describe the drought risks, impacts, and factors increasing or decreasing the likelihood of impactful drought over time in the study area. The interviewees were randomly chosen. Some were more in-depth interviews that lasted an hour, in other cases a short conversation, depending on the person's availability. All the interviewees provided consent before being interviewed. The interviews were not recorded, but fieldwork notes were either written up while the interview was ongoing or written up immediately afterwards. Fieldwork notes were transcribed and analyzed using Atlas.ti software to identify key themes and patterns related to drought impacts and public policies.

**2.2.3. Policy documents data**

Policy documents specifically related to supporting farmers, their families, and rural communities were collected to gain a complete understanding of the objectives, strategies, and implementation frameworks of these relevant policies and programs within the study area. They were analyzed for descriptive information and coded for key elements such as goals, instruments, and responsible organizations.

One limitation is that policy documents may not always accurately reflect the actual implementation or impact of a policy. To overcome this limitation, we also used our fieldwork experience and interviews to understand the nuances about the implementation of policies and their influences on livelihoods on a local level.

**2.2.4. Analysis and synthesis**

We used deductive reasoning to categorize the three types of impacts of drought. With this framework, we started to elaborate the different cascades in relation to the most common impacts recognized in our field work campaigns. The relationship between drought impacts and policy responses was explored to elucidate how policies evolve to mitigate the cascading effects of drought.

# 3. Results

Figure 3 presents 1,933 reported impacts*, categorized into three main types: hydrological, agricultural, and socio-environmental-economic impacts of drought. These impacts were reported in a open question, where observers were given the freedom to express what they considered relevant during that period. Consequently, the reported impacts were not limited to drought alone, for instance, impacts related to pests and socioeconomic impacts were reported, that may or not have been aggravated (or alleviated) by drought.

The bars reveals that most impacts are linked to hydrological drought (N =1.187), with agricultural drought (N = 718) being the next most common. In contrast, socio-environmental-economic impacts of drought (N = 28) exhibit the lowest frequency. Due to scale considerations, this bar has its details unclear. These impacts include: wildfires (8), high production costs (9), and socioeconomic impacts* (11).

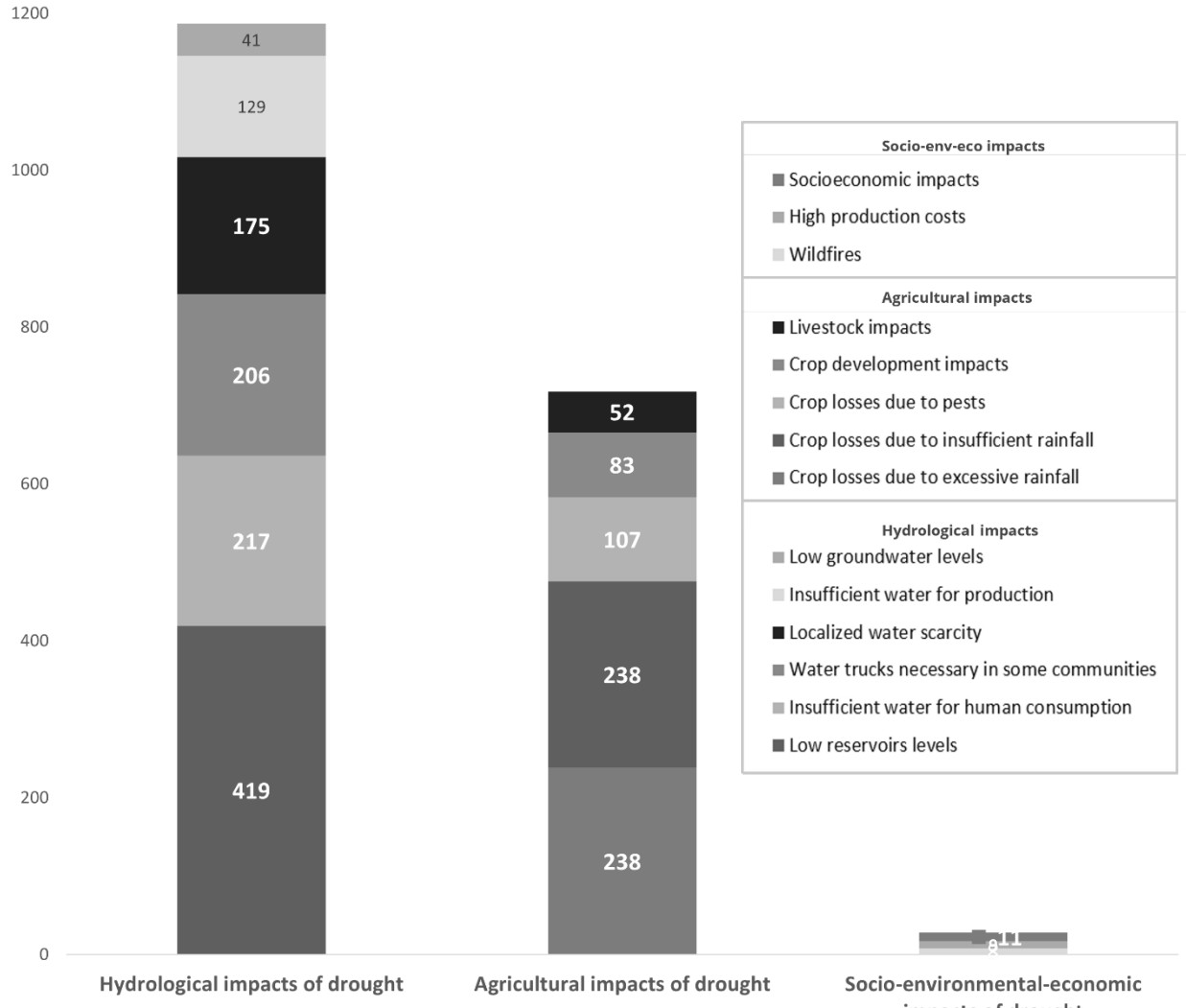

Figure 3: Number of impacts reported by observers in relation to categories of drought

* "Socioeconomic impacts" combines rare examples of impact codes such as "loss of income" (one occurrence), "migration to urban areas" (one occurrence), and "livestock farmers suffering" (four occurrences), "reduced economy" (one occurrence), "social impacts" (one occurrence), "worrying situation" (three occurrences).

An intriguing finding was the equal frequency of impacts (238) related to crop losses caused by excessive rainfall and insufficient rainfall. Subsequent interviews with observers confirmed excessive rainfall caused losses, including both waterlogging of low-lying areas and untimely rains during harvest. This refers to high-intensity rainfall or excessive rainfall at unexpected times, rather

than simply high volumes. From our interviews, we learned that beans are the crop type most
impacted by excessive water.
Drought impacts cascade in various directions. In Figure 4, we illustrate several potential directions
based on the main impacts reported in the observer's monthly questionnaires (Figure 3), and
complemented with fieldwork notes about the cascade in a local level. This is a simplification of a
full range of cascading impacts.

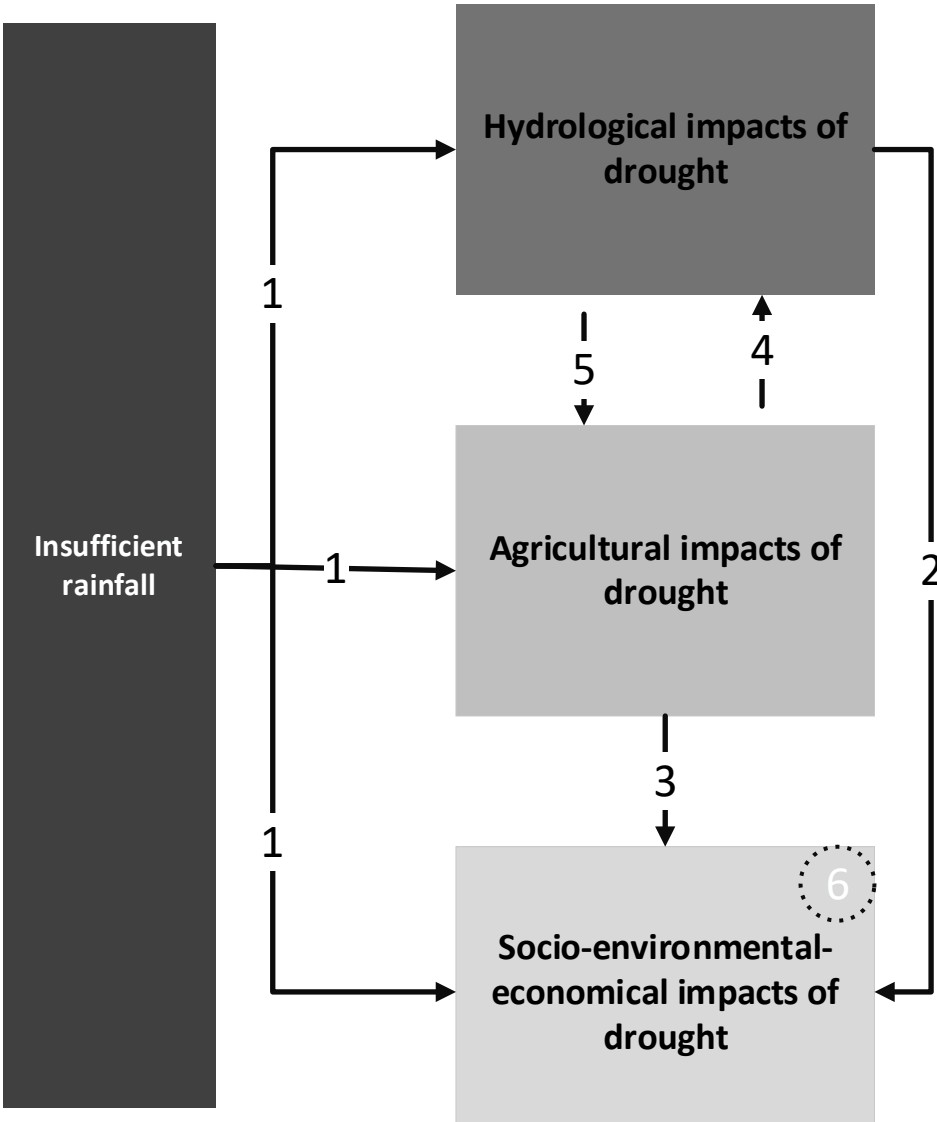


Figure 4: Schematic illustration depicting various directions for the cascading of drought impacts.
**Cascade direction number 1, insufficient rainfall causing hydrological, agricultural or socio-**
**environmental-economical impacts of drought.** This direction can follow various paths, directly
extending from the primary driver to others. Insufficient rainfall can lead to localized water
scarcity, crop development impacts, and increased risk of wildfires. e.g. Municipalities have
indicated that the absence of rainfall, coupled with low air humidity and elevated temperatures, has
led to fires in certain regions.
**Cascade direction number 2, hydrological impacts of drought to socio-environmental-**
**economical impacts of drought.** Low water availability for human consumption may have
socioeconomic consequences, necessitating expenses on water trucks to fulfill household
requirements. e.g. In cases of reduced rainwater cistern levels designated for human consumption
or low reservoir levels, households turn to buying water from water trucks to replenish their
cisterns.
**Cascade direction number 3, agricultural impacts of drought to socio-environmental-**
**economical impacts of drought.** Deficiency in soil moisture can adversely affect the cultivation
of crops intended for feeding livestock, leading to the purchase of expensive animal feed. e.g. The
decrease of soil-moisture in the allocated floodplain area designated for the cultivation of grass and
sorghum for silage provision during dry periods has led to a consequential shift in agricultural
practices. Consequently, some farmers have decided to sell a portion of their livestock, thereby
reducing the size of their herds, enabling farmers to buy extra complements like soybeans to feed
their animals.
**Cascade direction number 4, agricultural impacts of drought leading to hydrological impacts**
**of drought.** Reduced soil moisture for crop development requires irrigation, prompting individuals
to draw water from reservoirs, potentially resulting in diminished reservoir levels. e.g. In 2015, the
Pirabibu reservoir's water level dropped to zero due to human pressure on irrigation for producing
forage during a period of precipitation deficit. This process is explained in details by (Kchouk et
al., 2023).
**Cascade direction number 5, hydrological impacts of drought leading to agricultural impacts**
**of drought.** The hydrological impacts of low reservoir levels (resulting in reduced stream flow and
low groundwater levels) and soil-moisture drought can affect the growth of crops and insufficient
water available for irrigation.e.g. Such conditions are applicable to maize cultivation, where
insufficient soil moisture hinders fruit development, leading to a deceleration in the growth of the
plant.
**Cascade direction number 6, whithin the socio-environmental-economic impacts of drought**.
These impacts are interconnected with the livelihoods and well-being of farmers and households.
e.g. Agricultural losses, which result in less feed for animals, cascading into an extra expense by
having to buy feed in the market, consequently there is a reduced margin and loss of income.
Therefore, farmers will sell their assets - the livestock -, resulting in more loss of income.
Figure 4 demonstrates how policy responses intervene to alleviate the ongoing drought impacts
cascade. To enhance readability and comprehension, policies are represented with pictures taken
during fieldwork. Table 3 provides a comprehensive overview of these policies, detailing the
specific issues addressed by them, and the type of instruments employed to address each problem,
the year of enactment, and the managing organization.

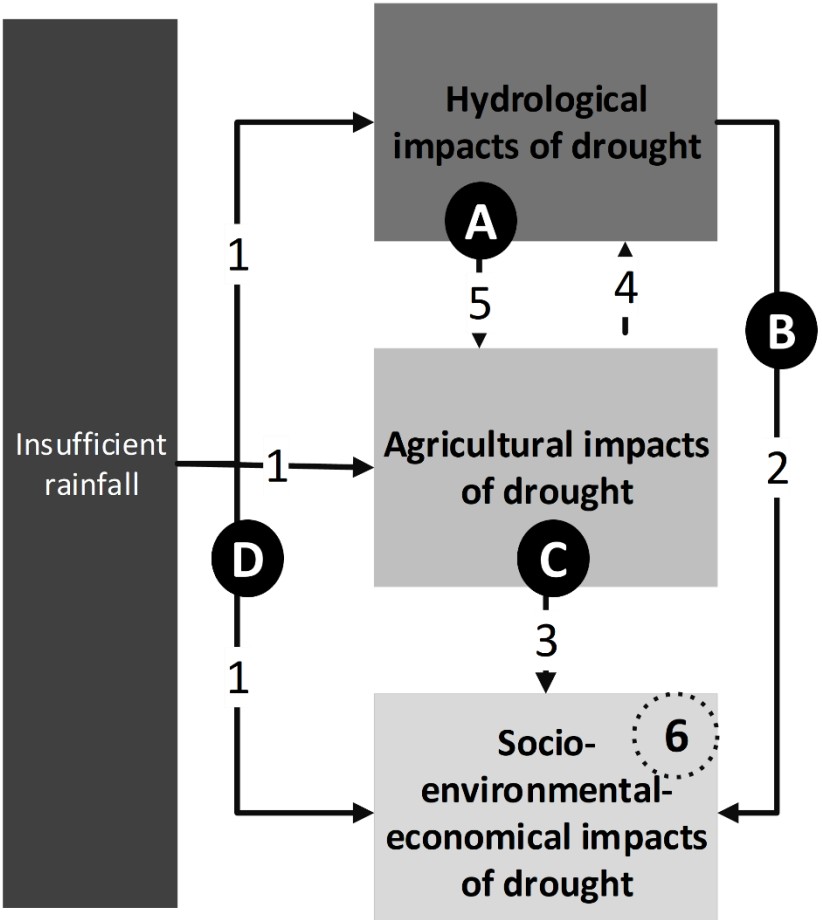


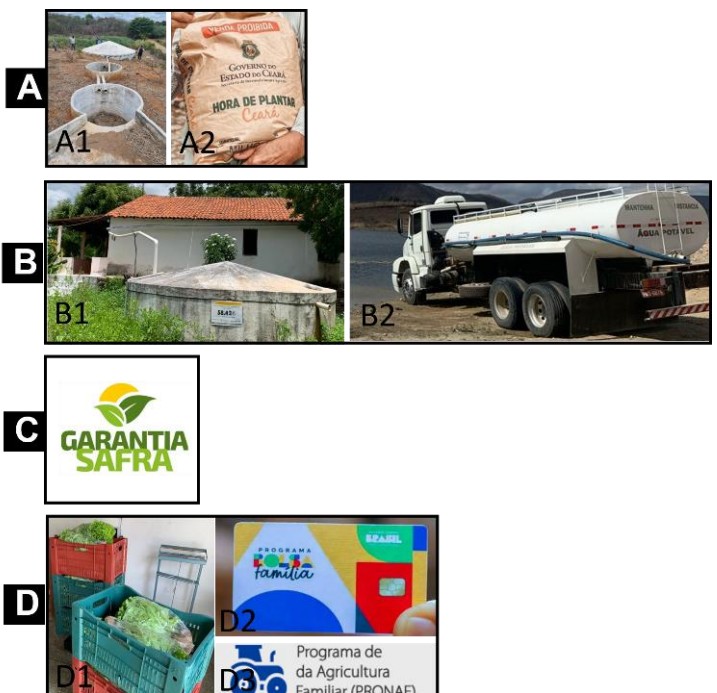

Figure 5: Policy responses and their role in alleviating the cascade of drought impacts.

**Policy response path A**, from hydrological drought impacts to agricultural impacts of drought, two policies are implemented: One is the *production cistern* (A1), 52m$^3$ rainwater harvest reservoirs to water livestock and provide small-scale irrigation to guarantee food security for the household. Another is *Hora de Plantar* (A2), the distribution of beans and maize seeds with high genetic potential for drought resistance to smallholder farmers prior to the rainy season. This is the only policy specific to Ceará state; all others are Nacional policies.

**Policy response path B**, from hydrological to socio-environmental-economical impacts of drought, two policies are represented: One is the *cistern* (B1) with 16m$^3$ rainwater harvest reservoirs for domestic purposes. Second is *water trucks* (B2) for distributing water for domestic uses, coordinated by the federal government in collaboration with the Ministry of National Integration and the Ministry of Defense.

**Policy response path C,** addressing the agricultural and socio-environmental-economic impacts of drought, is *Garantia Safra* (C), a cash transfer insurance that provides payments to farmers facing 50% or more crop losses due to drought or excess water. Smallholders are required to enroll in the program annually within their municipality. The financial responsibility for *Garantia Safra* is distributed among farmers, municipal administrations, federal states, and the federal government. This policy response helps mitigate the socio-economic consequences of crop losses by providing financial support to affected farmers, thereby preventing further cascading impacts.

**Policy response path D**, from insufficient rainfall to socio-environmental-economic impacts of drought. One policy is *Bolsa Família* (D2), a conditional cash transfer (~ \$120) for families.

Conditionalities are ensuring that children attend school and receive necessary vaccinations.
Second, are the Food Acquisition Program (PAA) and the National School Feeding Program
(PNAE) (D1), are public procurement policies that promote social inclusion and poverty reduction
by connecting small-scale farmers with government institutions that procure food. Regarding PAA,
one farmer mentioned, "*the [PAA]... during the dry season is helpful because it is guaranteed, and*
*you can already count on the resources*." Third is the National Program for Strengthening Family
Agriculture (PRONAF) (D3). Access to rural credit enables farmers to obtain financing under
favorable conditions tailored to their needs and interests.
Table 2: Policies implemented in the region

| Policy response* | Problem the policy address | Instrument | Year of enactment | Managing organization |
|---|---|---|---|---|
| **National Program for Strengthening Family Agriculture (PRONAF)** | Lack of access to credit by smallholders | Rural credit lines | 1995 | Ministry of Agriculture, Livestock and Supply |
| **Crop Guarantee (*Garantia Safra*)** | Crop losses | Cash Transfer | 2002 | Ministry of Agrarian Development and Family Agriculture |
| **Family Allowance (*Bolsa Família*)** | Families in extreme poverty conditions | Cash transfer | 2003 | Ministry of Development and Social Assistance, Family and Fight Against Hunger |
| **Cisterns (*Programa 1 milhão de cisternas*)** | Water access for domestic uses | Rainwater harvest reservoirs | 2003 | Ministry of Social Development |
| **2nd water cisterns (P1+2)** | Water access for productive uses | Rainwater harvest reservoir | 2015 | Ministry of Social Development |
| **Food acquisition program (PAA)** | Insufficient market to smallholder farmer's | Public procurement contracts with smallholder farmers | 2003 | Ministry of Social Development |
| **Food acquisition program for Schools (PNAE)** | Insufficient market to smallholder farmer's | Public procurement contracts with smallholder farmers to supply schools | 2010 | Ministry of Education |
| **Water trucks (*Operação Caminhão Pipa*)** | Lack of water for domestic purposes | Distribution of water | 2012 | Ministry of National Integration and the Ministry of Defense |
| **Time to plant (*Hora de Plantar*)** | Lack of drought resistant seeds and seedlings | Distribution of seeds and seedlings of high genetic potential to smallholder farmers | 1987 | Ceará State Level - Secretary of Agrarian Development |
| **Family Health Strategy (*Estratégia Saúde da Família*)** | Health Diseases | Basic health care (doctors, nurses, hospitals) | 1991 | Ministry of Health |

*Policy response translated to English with the equivalent name in Portuguese in brackets
The data analysis indicates that socio-environmental-economic impacts have the lowest frequency
of reporting, suggesting that public policies may have been effective in alleviating some of the
cascade of impacts. However, this should not be generalized to environmental impacts, as farmers
and observers might not have reported on the state of ecosystems, including forests, freshwater
systems, and water quality in lakes and rivers. Additionally, no specific policies targeting the
alleviation of environmental impacts were identified. It's important to acknowledge that the low
frequency of reported socio-economic impacts may not necessarily indicate a reduction in these
impacts. Instead, it could reflect the way the data was collected or what people chose to focus on.
Tangible impacts, such as reduced crop yields, are often easier to notice and report compared to
less tangible or indirect impacts like migration or reduced income. Therefore, drawing conclusions
about the effectiveness of policies based solely on the frequency of mentions is limited. A more
robust analysis would require quantitative measurements of individual impacts or evidence that the
low number of mentions is not due to biases in data collection or reporting.
Fieldwork interviews revealed that droughts nowadays are less impactful because of the social
protection net that exists with programs like *Bolsa Família*. One interviewee mentioned that in the
1993 drought, she did not experience thirst, but this drought left a significant mark on her because
she was pregnant with her first daughter at the time. Food was scarce, and she had to resort to
eating a local bird, a low nutritious food they'd never eat if it wasn't an emergency. There was no
assistance from the government, she said "*with money, one could buy everything*". This drought's
main impacts were on food, water, and later finances. In contrast, she mentioned that during the
2012-2018 drought, fish died in the mud, and only one water truck (16 m$^3$) would come per month
for 20 families. This scarcity led to conflicts, albeit minor. They received crop insurance, and the
impacts were primarily related to water scarcity. Despite the 2012-2018 drought being statistically
more severe than the 1993 drought, the support of social programs made the impacts less severe.

## 377 4. Discussion

Our research findings indicate that policy responses play a crucial role in alleviating the cascade of
drought impacts, leading to variations in the distribution of these impacts depending on the extent
of local implementation. The reduction in the frequency and severity of impacts, particularly on
livelihoods, reflects the positive effects of development policies in fostering economic dynamism
within the region. Programs such as Bolsa Família, Garantia Safra, the Food Acquisition Program
(PAA), and the National School Feeding Program (PNAE) have been instrumental in 'breaking' the
cascade into socioeconomic impacts by providing crucial financial resources to vulnerable
populations, thus giving them the means to cope with drought. Stakeholders, including farmers and
observers, noted that recent drought periods (2012–2018) were more manageable compared to the
past (80s and 90s) when such governmental programs were absent. Today, droughts no longer
result in hunger and mass migration in the rural communities of the Brazilian semiarid region as
they once did. However, it is important to recognize that while these welfare programs have
significantly mitigated the immediate impacts of drought, they may have done so more by
providing temporary relief rather than by promoting long-term adaptation strategies. This suggests
that while the population is better equipped to manage droughts, they are not fully adapted to the
phenomenon, highlighting the need for a continued focus on sustainable adaptation measures
(Mancal et al., 2016).

However, while these programs have significantly contributed to 'breaking' the cascade of socioeconomic impacts by fostering long-term resilience, their effectiveness is contingent on integration with broader strategies aimed at sustainable development and climate adaptation. This concern is particularly relevant as climate change continues to transform the risks faced by individuals and households, potentially exacerbating poverty, inequality, and social instability. Therefore, while the population is better equipped to manage droughts, they are not fully adapted to the phenomenon, highlighting the need for an integrated approach where social protection is aligned with climate policy to strategically contribute to long-term resilience and well-being (Bedran-Martins et al., 2018). Although these programs have succeeded in improving material quality of life and increasing the Human Development Index, they do not fully address the multifaceted nature of vulnerability in the context of climate change. For example, despite the improvements in material conditions, the subjective well-being of households continues to be influenced by factors beyond economic security, such as health status and safety (Costella et al., 2023).

This perspective aligns with other research indicating that the insurance has transformed into a regular cash transfer linked to regular crop losses, serving more as financial support for household expenses than a cover for productive costs (Milhorance et al., 2020). However, while research suggests *Bolsa Família* has positively impacted income, it does nothing to address the risk of food insecurity during drought events. This indicates a 'poverty trap', where families continuously struggle with drought challenges without overcoming the underlying conditions that render them vulnerable (Lemos et al., 2016). During our study, we identified public procurement initiatives supporting family farming as a noteworthy case for overcoming this 'poverty trap'. Families exhibited greater resilience to drought-related challenges due to increased income, enabling them to enhance and diversify their production to PAA and PNAE another (Kchouk et al., n.d.). Those are not reactive policies aiming to address one cascaded impact, but rather to stop the cascade from agricultural impacts to impacts on livelihood, creating economic stability for families that diversified their production. The PAA offers access to a stable market and increases farmers' income by providing a reliable market for their produce (Mesquita and Milhorance, 2019).

Many reports emphasize ongoing water supply challenges, indicating that despite the construction of reservoirs, the hydrological drought continues to pose significant challenges in the semiarid region.While policies prioritized the extensive construction of reservoirs to enhance water supply (Cavalcante et al., 2022), the persistent issue of water access remains (Gutiérrez et al., 2014). This exhibits signs of maladaptation, leading to increased water consumption and insufficient water redistribution among regions (Machado and La Rovere, 2018). As a result, these are indications of unintended consequences due to the interactions between human activities, water infrastructure, and natural systems (Di Baldassarre et al., 2018; Ribeiro Neto et al., 2022). Another illustration of this is the promotion of non-adapted crops for the region, such as the cultivation of rice. This highlights a pattern wherein reliance on water resources has increased gradually due to incentives for economic activities not aligned with the region's environmental conditions.

The distribution of water trucks by the policy '*operação carro pipa*' depends on the severity of
drought. When municipalities declare an 'emergency situation', it is legal recognition by the
affected municipality of an exceptional situation caused by a disaster. However, our qualitative
data showed that there are other ways in which water trucks are distributed. Some municipalities
operate their own water trucks, allowing them to make independent decisions regarding water
usage. Farmers also have the option to directly purchase private water trucks. Our interpretation
for the high number of impacts reported as 'water trucks necessary in some communities (n=206)'
is that they cannot always be considered impacts. Rather, it reflects an ongoing regional dynamic
– water trucks are part of the water system to avoid the cascade into water shortage – that persists
regardless of the formal classification of the period as a severe (or weak or no) drought (Walker et
al., 2024).
The focus on environmental drought was to highlight the interconnectedness of natural and human
systems (Srivastava and Maity, 2023). The experiences of those directly affected by drought in
Northeast Brazil offered powerful insights into the real-world impacts of this phenomenon,
revealing that drought extends far beyond water scarcity. While the more visible effects, like
reduced crop yields, are often easier to notice and report, the less tangible or indirect impacts on
ecosystems frequently go unaddressed. Future studies should aim to bridge this gap by specifically
examining ecological drought impacts, and how it afftects biodiversity, allowing for a better
understanding of how these impacts are distributed on ecosystems.
A cascade typically refers to a series of events or processes linked and often result in a chain
reaction. Our approach looked at individual alleviation of drought impacts. It is a simplification
representing the cascade as a linear process following from agricultural impacts of drought
progressing to hydrological to socio-environmental-economic impacts or just connecting one type
of drought impacts to the other. A recent study investigating the 2018/2019 drought in Germany
used sequential pattern mining analysis, revealing that the impacts exhibited simultaneous and
distinct patterns (de Brito, 2021). It is worth mentioning that we attempted to analyze our dataset
using the same methods to identify patterns of cascade of drought impacts. However, the limited
quantity of data proved insufficient for the machine to detect patterns. This could have helped us
recognize additional types of connections between patterns that may not have been apparent
through human reasoning alone. Despite our efforts and endorsement for this analysis approach,
our attempt was unsuccessful due to methodological limitations.
While studies using machine learning to study drought impacts represent a notable advancement
in drought management, we advocate for the integration of social and qualitative data to gather the
perspectives of people "on-the-ground" who directly experience the impacts. This is crucial,
because the collective capacity of stakeholders across different scales (spatial, jurisdictional, and
temporal) determines whether a system adapts, collapses, or shifts in response to drought (Kchouk
et al., 2023). Artificial intelligence does not yet capture these nuances of adaptation or impacts that
only experience of local context can provide. Our study, which leverages data from traditionally
low-data environments, highlights the importance of integrating and validating these often-

overlooked sources. This approach enriches our understanding of drought dynamics, particularly in vulnerable regions, highlighting how such data can reveal the nuanced impacts of drought on smallholders. These smallholders are among the most vulnerable to climate extremes, and their experiences provide valuable insights into how policy measures can better support long-term resilience.

Our recommendation for practice is to invest in climate adaptation projects within the region. It is noteworthy that the region receives less research and financial attention from both the government and international donors than the Amazon region (Santos et al., 2011). We suggest the promotion of local crop varieties and adapted breeds aligned with the cope-with-drought approach (Cavalcante et al., 2022) and the consideration of local practices that have achieved a sustainable balance between their livestock and milk production, enabling them to thrive even during prolonged drought periods (Kchouk et al., 2023). We also propose implementing policies that enhance ecosystem services, such as soil conservation and water retention through agroforestry practices, to further alleviate residual drought impacts in the semiarid region.

In response to Michel Jarraud's claim, our investigation revealed that policy responses have been somewhat effective in alleviating the socioeconomic impacts due to the development policies in place. However, drought-related policies still tend to be reactive, such as implementing crop insurance only after drought impacts have occurred. This reactive approach presents an opportunity for improvement. For instance, moving towards proactive policies, such as utilizing cash transfers based on forecasted impacts rather than responding to those that have already happened, could enhance the effectiveness of drought management. This shift emphasizes the importance of forecasting and associated proactive measures. After several years of research and discussion on drought, we advocate that drought should be managed as a cross-cutting issue that impacts multiple sectors simultaneously, necessitating a comprehensive and interconnected approach. Drought's far-reaching impacts go beyond water scarcity, influencing agricultural production, socio-economics, and increasing the risk of fires. Therefore, we highlight the significant role that public policies can play in mitigating the cascading effects of drought, particularly those impacts not directly related to increasing water availability.

This analysis opens space for further research in other regions of the world where drought impacts are also monitored, for instance the USA[1] and Central and Eastern Europe[2]. This type of analysis should be conducted to assess policy effectiveness to deal with drought impacts, and that can only be done with continual drought impacts monitoring, which is unfortunately lacking in most of the world (Smith et al., 2023). Another avenue for further investigation lies in longitudinal studies covering extended periods, encompassing periods characterized by both drought and rainfall occurrences, thereby comprehensively addressing diverse hydrological circumstances. This aspect,

---

[1] https://droughtmonitor.unl.edu/
[2] https://questionnaire.intersucho.cz/en/

which was not explicitly delineated in the present study, represents a crucial limitation that requires
attention in future research.

## 5. Conclusion

This study aimed to understand the role of society in mitigating drought impacts, particularly
through policy responses. Among the least frequently reported impacts were those pertaining to
the socio-environmental-economic aspects of drought, particularly affecting livelihoods. Most
impacts were hydrological, suggesting that the construction of reservoirs may not have adequately
addressed the challenges posed by the semiarid region, leading to unintended consequences of
overreliance on these reservoirs.
Despite the positive impacts of public policies that stimulate economic activity within the region,
persistent socioeconomic impacts of drought persist. Therefore, we emphasize the significant
contribution of public policies to mitigating the cascading effects of drought impacts that extend
beyond simply increasing water availability.
Our analysis also highlights a tendency towards reactive rather than proactive policy measures, as
evidenced by the implementation of crop insurance initiatives post-event. This suggests a need for
more proactive policy approaches to drought management, evidenced in our research by the
distribution of beans and maize seeds with high genetic potential for drought resistance to
smallholder farmers prior to the rainy season.
For drought management, we recommend that drought should be managed as a cross-cutting issue
that affects and is relevant to multiple sectors simultaneously, necessitating a comprehensive and
interconnected approach to its understanding and addressing. We also raise attention to the limited
number of adaptation projects within the semiarid region, and a lack of financial and research
support compared to more prominent regions in Brazil, such as the Amazon. For future research,
we advocate for the integration of social and qualitative data alongside machine learning
approaches to comprehensively capture the nuanced dynamics of drought impacts and adaptation
strategies.

## Author contributions

LC and DWW initiated the original idea and conceptualized the research in collaboration with SK
and GRN. LC, SK, GRN and DWW conducted field work interviews and analysis of data. DWW
analyzed the impacts monitoring data with support of LC, SK and GRN. LC analyzed the policy
documents data. TMNC and MMB performed natural language processing and analysis. The
research was supervised by WP, DWW, AD and PvO. PvO acquired financial support for the
project leading to this publication. All co-authors contributed to the interpretation of the results and
to the article writing.

# Funding

This research was supported by the Dutch Research Council (NWO) and the Interdisciplinary Research and Education Fund (INREF) of Wageningen University, the Netherlands (grant no. W07.30318.016).

# Competing interests

The contact author has declared that none of the authors has any competing interests.

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
