# Peer review of "From insufficient rainfall to livelihoods: understanding the cascade of 1 drought impacts and policy implications 2"

_EGUsphere, 2024_

## Author Comment (AC1)

**Response to reviewer number 1 – NHESS SPECIAL ISSUE Drought, society, and ecosystems (NHESS/BG/GC/HESS inter-journal SI)**

Dear Anastasya Shyrokaya,

Thank you for your thoughtful and constructive comments on our manuscript, for your time reviewing it and for your prompt review. Your feedback has contributed to improving the clarity and depth of our work. I apologize for the delay in my response, as I needed some time away from research before my PhD defense. As the first author, I am now able to fully address your comments. Please, see them attached.

Kind Regards,

Louise Cavalcante on behalf of all co-authors.

| Comment | Response |
|---|---|
| Line 53-65: This paragraph introduces different types of droughts. I wonder why environmental/ecosystem type of drought is not included among meteorological, agricultural and hydrological types, which is still a physical manifestation of impact before introducing the socio-economic drought? (pls refer to AghaKouchak et al. 2021). If the reason for this is the limited number of impact reports for environmental drought (as shown in Wildfires category in S1 Table 3), leading to its merging with socio-ecological-economic impact category, I would suggest first introducing the environmental drought and its physical impacts, and then explaining why they were merged. | We appreciate this suggestion and have revised the introduction to include a discussion of environmental/ecological drought. We also referred to AghaKouchak et al. 2021.

 We now introduce environmental drought and its physical impacts before discussing socio-economic drought. This revision aligns with a similar request made by another reviewer, further reinforcing its importance.

 "Another category, often considered by ecologists, is environmental or ecosystem drought, which refers to a temporary shortfall in water availability that pushes ecosystems beyond their vulnerability limits, disrupts ecosystem services, and triggers feedback loops within both natural and human systems (Crausbay et al., 2017)."

 We have reviewed the paragraphs on drought definitions, and they now read as follows:

[revised manuscript text omitted]

 The state has various economic activities, with key sectors including industry, particularly textiles and automotive manufacturing, as well as tourism driven by its tropical beaches and wind sports. |
| Table 1: I would leave a space before "July 2019, November and December 2021 and April 2022" to make it more clear that it indicates the period of Field work data. | Done! This was also asked by the other reviewer. |
| Line 192-193: "Policy documents were collected to understand the objectives and strategies of relevant policies and programs in the study area". I would advise specifying more what is meant by "relevant". E.g. policies related to supporting farmers and their families etc. | Done!

 Policy documents specifically related to supporting farmers, their families, and rural communities were collected to gain a complete understanding of the objectives, strategies, and implementation frameworks of these relevant policies and programs within the study area. |
| "*These impacts include: (8), high production costs (9), and socioeconomic impacts* (11)*". Is there an impact type missing in front of (8)? Should be "Wildfires" based on Fig.2? | I had forgotten to write this, or it might have been accidentally deleted. Thank you for noticing.

 "These impacts include: wildfires (8), high production costs (9), and socioeconomic impacts* (11). " |
| Figure 2: I would add "impacts" to socio-env-eco and start with the capital letter "Socio-env-eco". Could potentially move these titles to the | Done! |

| | |
|---|---|
| top within each box and place them horizontally. | |
| Figure 3: It's a bit difficult to read the sentence "Socio-environmental-economical impacts of drought" within the last box, would recommend making the contrast more visible. Same for Figure 6 as Figure 3 is part of it. | Done! |
| Same for Figure 6 as Figure 3 is part of it. | Done! |
| Figure 3: "illustration illustrating" in the Figure's caption. | Done, now reads as: Schematic illustration depicting various directions for the cascading of drought impacts. |
| Line 308-310: "*The data analysis indicates that socio-environmental-economic impacts have the lowest frequency of reporting, suggesting that public policies have been effective in alleviating the cascade of impacts*". I would not generalize this conclusion for environmental impacts – the farmers/observers might not report on the state of ecosystems incl forest, freshwater ecosystem, water quality in lakes/rivers etc. Also, there were no policies mentioned that were alleviating specifically environmental impacts. | We have revised this section to avoid generalization. Now read as: The data analysis indicates that socio-enviromental-economic impacts have the lowest frequency of reporting, suggesting that public policies may have been effective in alleviating some of the cascade of impacts. However, this should not be generalized to environmental impacts, as farmers and observers might not have reported on the state of ecosystems, including forests, freshwater systems, and water quality in lakes and rivers. Additionally, no specific policies targeting the alleviation of environmental impacts were not found. |
| Line 413: "*On drought related policies, they remain reactive, such as the crop insurance implement after drought impacts are experienced*". This presents an opportunity to mention an example of moving from reactive to proactive policies by e.g. using cash transfers based on forecasted impacts rather than responding to those that have already occurred, highlighting the importance of forecasting and associated proactive drought management. | We have incorporated this suggestion into the discussion, highlighting the importance of proactive policies and the potential benefits of using forecasted impacts to inform drought management strategies.Thanks for your thoughts on it. " In response to Michel Jarraud's claim, our investigation revealed that policy responses have been somewhat effective in alleviating the socioeconomic impacts due to the development policies in place. However, drought-related policies still tend to be reactive, such as implementing crop insurance only after drought impacts have occurred. This reactive approach |

| | |
|---|---|
| | presents an opportunity for improvement. For instance, moving towards proactive policies, such as utilizing cash transfers based on forecasted impacts rather than responding to those that have already happened, could enhance the effectiveness of drought management. This shift emphasizes the importance of forecasting and associated proactive measures.

After several years of research and discussion on drought, we advocate that drought should be managed as a cross-cutting issue that impacts multiple sectors simultaneously, necessitating a comprehensive and interconnected approach. Drought's far-reaching impacts go beyond water scarcity, influencing agricultural production, socio-economics, and increasing the risk of fires. Therefore, we highlight the significant role that public policies can play in mitigating the cascading effects of drought, particularly those impacts not directly related to increasing water availability". |
| S1. Table 2: A little unclear with the headings: I'm guessing that "*Survey Questions*", "*Alleviating Factor*" shouldn't have a circle in front since they're the same level as the rest of the headings? | Addressed, thanks for noticing. |
| S1.3 Policy documents data "*The selected documents were about the public policies reported by both farmers and observers in the interviews*": have you considered checking other relevant documents that were not mentioned by farmers/observers, just generally from legislative repositories? In case there are some policy documents that potentially help alleviate prevailing hydrological impacts, but are not used by farmers for some reason? | Thank you for your suggestion regarding the consideration of additional policy documents from legislative repositories. I would like to assure you that throughout the five years of my PhD research, I conducted a thorough review of all relevant policy documents, including those not specifically mentioned by farmers or observers.

This extensive search included a comprehensive examination of legislative repositories to ensure that no significant policies were overlooked. The documents selected for the study represent the most pertinent policies related to drought mitigation and hydrological impacts in the region.

These were carefully chosen based on their relevance and practical application in the study area. |

---

## Author Comment (AC2)

Dear Dr. Kelly Smith,

Thank you for your thoughtful and constructive comments on our manuscript, as well as for the time and promptness of your review. Your feedback has greatly contributed to improving the clarity of our work, and several of your comments aligned with those from the other reviewer. I apologize for the delay in my response, as I needed some time away from my research before my PhD defense. As the first author, I am now able to fully address your comments below.

Kind Regards,

Louise on behalf of all co-authors.

| Comment | Response |
|---|---|
| Given the interconnected, cascading nature of different types of drought and drought impacts, it is perhaps inevitable that questions arise about how many distinctions is too many and how many is enough.

I recommend that the authors add ecological drought to the paragraph on types of drought – lines 53-65 – with a citation to work by Crausbay et al.:

Crausbay, S. D., Ramirez, A. R., Carter, S. L., Cross, M. S., Hall, K. R., Bathke, D. J., ... & Sanford, T. (2017). Defining ecological drought for the twenty-first century. Bulletin of the American Meteorological Society, 98(12), 2543-2550. https://doi.org/10.1175/BAMS-D-16-0292.1

Also, the statement on lines 53-54 isn't quite right. The different categories of drought relate more to context and discipline, not to characteristics such as duration, extent and intensity. Tsakiris is saying that drought indices that help measure those characteristics can work with each type of drought, not that indices are used to categorize drought.

As first mentioned on line 180, grouping "socio-environmental-economic" impacts of drought is quite a large catch-all category.

Also, here you are acknowledging environmental (ecological?) drought but it is grouped with socio-economic drought. That group has quite a span. It is possible that the main uniting feature of socio-economic and environmental impacts is that people | As you suggested, we have incorporated the concept of ecological drought into our manuscript. Specifically, we added a discussion of ecological drought in the paragraph on types of drought, citing the work of Crausbay et al. (2017) as you recommended. This addition helps to further emphasize the multidisciplinary nature of drought research and highlights the importance of understanding drought impacts on ecosystems.

Now it read as:

"Another category, often considered by ecologists, is environmental or ecosystem drought, which refers to a temporary shortfall in water availability that pushes ecosystems beyond their vulnerability limits, disrupts ecosystem services, and triggers feedback loops within both natural and human systems (Crausbay et al., 2017)."

We also integrated ecological drought into the discussion section, where we explore the broader implications of our findings.

"The focus on environmental drought was to highlight the interconnectedness of natural and human systems (Srivastava and Maity, 2023). The experiences of those directly affected by drought in Northeast Brazil offered powerful insights into the real-world impacts of this phenomenon, revealing that drought extends far beyond water scarcity. While the more visible effects, like reduced crop yields, are often easier to notice and report, the less tangible or indirect impacts on ecosystems frequently go unaddressed. Future studies should aim to bridge this gap by specifically examining ecological drought impacts, and how it afftects |

| | |
|---|---|
| are less likely to recognize and describe them. Please consider splitting environmental impacts from socio-economic, or at least providing a good explanation for why they are grouped that way. I'd consider socio-economic vulnerability to be quite distinct from environmental vulnerability.

A group this large and diverse increases the risk that you will miss or oversimplify causal pathways. | biodiversity, allowing for a better understanding of how these impacts are distributed on ecosystems" .

Regarding your suggestion to reconsider the grouping of environmental drought impacts with socio-economic drought, we have given this careful thought. While we understand the distinct nature of environmental (or ecological) impacts, we have chosen to maintain the categorization of environmental drought within the broader socio-economic context in this paper. The rationale for this decision lies in the specific impacts we are examining, particularly those related to wildfires. Wildfires are a significant concern that links environmental and human dimensions. Therefore, in the context of our study, these impacts are closely intertwined with human activities, justifying their inclusion in the socio-economic category. |
| Counting mentions

Lines 308-310: You can't conclude that successful policies are responsible for reducing the cascade of impacts based on frequency of mentions. To draw that conclusion you would need quantitative measurements of individual impacts, such as migration or reduced income, not the number of times that those impacts were mentioned in reports. … Or you would need evidence that the low number of mentions of socio-economic impacts isn't due to the way the data is collected or to what people pay attention to. Typically, tangible impacts such as reduced crop yield are much easier to notice and report than less tangible or less direct impacts. | In response to your feedback, we have revised the relevant section to clarify that while the frequency of mentions provides some insight into the perceived success of policies, it is not sufficient to draw definitive conclusions about their effectiveness.

Now read as: The data analysis indicates that socio-environmental-economic impacts have the lowest frequency of reporting, suggesting that public policies may have been effective in alleviating some of the cascade of impacts. However, this should not be generalized to environmental impacts, as farmers and observers might not have reported on the state of ecosystems, including forests, freshwater systems, and water quality in lakes and rivers. Additionally, no specific policies targeting the alleviation of environmental impacts were identified. It's important to acknowledge that the low frequency of reported socio-economic impacts may not necessarily indicate a reduction in these impacts. Instead, it could reflect the way the data was collected or what people chose to focus on. Tangible impacts, such as |

| | reduced crop yields, are often easier to notice and report compared to less tangible or indirect impacts like migration or reduced income. Therefore, drawing conclusions about the effectiveness of policies based solely on the frequency of mentions is limited. A more robust analysis would require quantitative measurements of individual impacts or evidence that the low number of mentions is not due to biases in data collection or reporting. |
|---|---|
| Lines 351-352: Again, be very careful about imputing any meaning to the quantity of hydro-related reports. It may be more relevant to cite the content, something along the lines of "many reports underscored water supply challenges remaining despite the construction of reservoirs." | Thank you for your comment. We understand the importance of not over-interpreting the quantity of hydro-related reports. Instead, we will emphasize the content of these reports.

Now it read as:
Many reports emphasize ongoing water supply challenges, indicating that despite the construction of reservoirs, the hydrological drought continues to pose significant challenges in the semiarid region. |
| Narrative strength

But the example from the interview starting on line 310 is good. You could make it stronger by more clearly delineating which experiences were from 1993 and which from 2012-2018. One way would be to add "In contrast" to the start of the sentence that begins on line 316. | Done.

In contrast, she mentioned that during the 2012-2018 drought, fish died in the mud, and only one water truck (16 m3) would come per month for 20 families. This scarcity led to conflicts, albeit minor. They received crop insurance, and the impacts were primarily related to water scarcity. Despite the 2012-2018 drought being statistically more severe than the 1993 drought, the support of social programs made the impacts less severe. |
| Discussion
If I am reading this correctly, it suggests that programs such as PAA and PNAE can block the cascade of impacts by giving farmers increased income. This is actually a key point for reducing societal vulnerability to drought – give people enough resources to have options. But it seems as though you may be giving too much weight to counter-arguments on lines 334-343, and/or you | To address your feedback, we have revised the discussion to more clearly articulate the dual role of these programs in both providing immediate relief and contributing to long-term resilience. We've also incorporated new citations to support this argument and to reconcile the potentially conflicting views on the effectiveness of these programs. |

| | |
|---|---|
| could do more to reconcile two conflicting views.

I couldn't tell from this article whether the people arguing that assistance doesn't solve the underlying problem believe that is true for all forms of cash transfer programs or for specific programs, or whether they are defining the problem in a way that doesn't separately consider human well-being. | Now it read as follows:

Our research findings indicate that policy responses play a crucial role in alleviating the cascade of drought impacts, leading to variations in the distribution of these impacts depending on the extent of local implementation. The reduction in the frequency and severity of impacts, particularly on livelihoods, reflects the positive effects of development policies in fostering economic dynamism within the region. Programs such as Bolsa Família, Garantia Safra, the Food Acquisition Program (PAA), and the National School Feeding Program (PNAE) have been instrumental in 'breaking' the cascade into socioeconomic impacts by providing crucial financial resources to vulnerable populations, thus giving them the means to cope with drought. Stakeholders, including farmers and observers, noted that recent drought periods (2012–2018) were more manageable compared to the past (80s and 90s) when such governmental programs were absent. Today, droughts no longer result in hunger and mass migration in the rural communities of the Brazilian semiarid region as they once did. However, it is important to recognize that while these welfare programs have significantly mitigated the immediate impacts of drought, they may have done so more by providing temporary relief rather than by promoting long-term adaptation strategies. This suggests that while the population is better equipped to manage droughts, they are not fully adapted to the phenomenon, highlighting the need for a continued focus on sustainable adaptation measures (Mancal et al., 2016).

However, while these programs have significantly contributed to 'breaking' the cascade of socioeconomic impacts by fostering long-term resilience, their effectiveness is contingent on integration with broader strategies aimed at sustainable development and climate adaptation. This concern is particularly relevant as climate change continues to |

| | |
|---|---|
| | transform the risks faced by individuals and households, potentially exacerbating poverty, inequality, and social instability. Therefore, while the population is better equipped to manage droughts, they are not fully adapted to the phenomenon, highlighting the need for an integrated approach where social protection is aligned with climate policy to strategically contribute to long-term resilience and well-being (Bedran-Martins et al., 2017). Although these programs have succeeded in improving material quality of life and increasing the Human Development Index, they do not fully address the multifaceted nature of vulnerability in the context of climate change. For example, despite the improvements in material conditions, the subjective well-being of households continues to be influenced by factors beyond economic security, such as health status and safety (Costella et al., 2023). |
| Lines 362-372: This is a good paragraph, pulling out some of the nuance in the data. | Thanks 😊 |
| Lines 373-382: This paragraph acknowledges some of the unresolved complexity, but please consider deleting it. Instead, lean into the excellent narrative accounts that you have collected. As you note beginning on line 395, you don't need artificial intelligence to understand what's happening to people on the ground. You have done a good job of listening to them and giving them a voice. | We have deleted this paragraph and added some comments about the strengthen of the data collected. Thanks for your comments on this.

Now it read as: Our study, which leverages data from traditionally low-data environments, highlights the importance of integrating and validating these often-overlooked sources. This approach enriches our understanding of drought dynamics, particularly in vulnerable regions, highlighting how such data can reveal the nuanced impacts of drought on smallholders. These smallholders are among the most vulnerable to climate extremes, and their experiences provide |

| | valuable insights into how policy measures can better support long-term resilience. |
|---|---|
| Line 409: What are natural values? | We recognized that this term could be vague and unclear. To address this, we have revised the sentence to:

We also propose implementing policies that enhance ecosystem services, such as soil conservation and water retention through agroforestry practices, to further alleviate residual drought impacts in the semiarid region. |
| Line 154: On Table 1, the delineation between the period of data collection for drought impacts monitoring data and for field work data is unclear. | Done, the other reviewer also made this comment. |
| Please review capitalization and punctuation around the use of "e.g." | Done, thanks for noticing it. |

FYI:  have also included a map, as asked by another reviewer.

---

## Author Response (AR2)

**Brasília, January 30ᵗʰ, 2025**

Dear Editors,

We sincerely appreciate your thoughtful suggestions, which have enhanced the quality of our manuscript. We fully agree that the figures and tables should be self-explanatory. Below, we provide detailed responses to each point raised by you. All changes have been incorporated into the revised manuscript.

Comment 1: *Your figure captions and table headers are exceptionally short and should be made such that your figures and tables are self-standing.* Response: Thank you for this suggestion. We have revised all figure captions and table headers to include more detailed descriptions, ensuring that each figure and table is self-standing and provides sufficient context for readers.

Comment 2: *The legend, labels, and grid in the map of Figure 1 are hardly readable.* Response: We appreciate this observation. We have revised Figure 1 to improve readability by enlarging the font size of the legend and labels, and adjusting the grid layout for better clarity.

**We addressed other textual comments as follows:**

**Original:**
*"we used a global rare dataset of continuously drought monitoring"*
**Revised:**
*"we used a globally rare dataset of continuous drought monitoring spanning over a decade"*

- Changed "global rare dataset of continuously drought monitoring" to "globally rare dataset of continuous drought monitoring" for better grammar and flow.

- Added "encompassing 3.5 years (February 2019 to October 2022)" to highlight the significance and time of the dataset.

**Final version:** Conducting a case study in Ceará state, northeast Brazil, we used a globally rare dataset of continuous drought monitoring encompassing 3.5 years (February 2019 to October 2022), complemented by interviews with smallholder farmers and agricultural extension technicians.

**Original:**
*"Most impacts are associated with hydrological impacts of drought"*
**Revised:**
*"Most reported impacts are associated with hydrological drought"*

- Reworded to improve readability and remove redundancy ("impacts" repeated twice).

**Final version:** Most reported impacts are associated with hydrological drought, revealing unintended consequences of investments aimed at increasing water supply.

We have also performed a thorough final read-through of the manuscript to ensure consistency, clarity, and accuracy. Additionally, our native English-speaking co-author has carefully reviewed the text to enhance its readability and correctness. We are confident that the revisions address all the comments provided, and we look forward to your feedback on the updated manuscript.

Thank you again for your guidance and support during the review process.

Best regards,
Louise Cavalcante and co-authors